# In-sensor computing using a MoS$_2$ photodetector with programmable spectral responsivity

Dohyun Kwak[1,2], Dmitry K. Polyushkin ®[1,2] & Thomas Mueller ®[1] ✉

Optical spectroscopy is an indispensable technique in almost all areas of scientific research and industrial applications. After its acquisition, an optical spectrum is usually further processed using a mathematical algorithm to classify or quantify the measurement results. Here we present the design and realization of a smart photodetector that provides such information directly without the need to explicitly record a spectrum. This is achieved by tailoring the spectral responsivity of the device to a specific purpose. In-sensor computation is performed at the lowest possible level of the sensor system hierarchy – the physical level of photon detection – and does not require any external processing of the measurement data. The device can be programmed to cover different types of spectral regression or classification tasks. We present the analysis of spectral mixtures as an example, but the scheme can also be applied to any other algorithm that can be represented by a linear operator. Our prototype physical implementation utilizes an ensemble of optical cavity-enhanced MoS$_2$ photodetectors with different center wavelengths and individually adjustable peak responsivities. This spectroscopy method represents a significant advance in miniaturized and energy-efficient optical sensing.

With the proliferation of sensing devices in a wide range of portable applications, from environmental monitoring to medical diagnostics and food quality control, the demand for more compact and energy-efficient solutions is increasing. In-sensor computing is an emerging field that aims to bring computation and data processing capabilities to the edge of a sensing application. It leverages the processing power of computing devices that are directly integrated into the sensors. This approach differs from traditional solutions, where the collected signals are usually transferred to an external, centralized system for further processing. Recently, analog-domain in-sensor computing has gained momentum[1,2]. There, the sensory information is processed in its raw, analog format, bypassing the need for analog-to-digital conversion. As a result, data processing is faster and more energy-efficient, since it eliminates the transfer and conversion steps. For example, the classification of optical images directly during the light detection process has been utilized in

several recent works[3–10]. Similarly, the acquisition of an optical spectrum is not an end in itself, but rather serves to extract certain information about a specimen under examination, whether for classification or quantification purposes.

In optical spectroscopy, reconstructive (or computational) spectrometers[11–20] have emerged as a promising alternative to their conventional grating- or interferometer-based counterparts. The underlying concept of these instruments is that superpositions of intensities at different wavelengths are recorded, instead of the usual measurement of spectral intensities at individual wavelengths. This can be achieved with an array of filters or, preferably, a photodetector with a variable spectral response. The input spectrum can then be reconstructed from a series of such measurements. These devices do not require bulky and fragile optical components, such as diffraction gratings or moving mirrors, and can therefore be integrated on-chip. However, although reconstructive spectrometers offer better

[1]Vienna University of Technology, Institute of Photonics, Gußhausstraße 27-29, 1040 Vienna, Austria. [2]These authors contributed equally: Dohyun Kwak, Dmitry K. Polyushkin. ✉e-mail: thomas.mueller@tuwien.ac.at

scalability than conventional solutions, they require considerable computational resources to reconstruct and analyze the optical spectra. Therefore, to further reduce energy consumption, latency, footprint, and cost, it would be desirable to move the processing of the highly redundant and unstructured sensory data entirely into the photosensors themselves.

The ambition of this work is to realize a photodetector that is able to provide such information without having to resort to any external data processing. To accomplish this, the device's spectral responsivity is adjusted to match a specific purpose. In-sensor computation occurs at the lowest level of the sensor system hierarchy, which is the physical level of photon detection. Our approach aims to improve the capabilities of optical spectroscopy by providing a compressed representation of the relevant information contained in an optical spectrum and represents a significant advance in miniaturized and energy-efficient optical sensing.

## Results

In Fig. 1, we compare the detection scheme presented in this article to a conventional optical sensing approach. Conventionally (Fig. 1a), incident light is dispersed to produce a spectrum and its corresponding vector representation, $\mathbf{p}$, is recorded. The digitized data are then processed using an external processing unit/computer, which utilizes a statistical or machine-learning algorithm to analyze $\mathbf{p}$ and extract the desired information, $\alpha_i$, from the measurement. In contrast, in our in-sensor computing scheme (Fig. 1b), the same algorithm is implemented in the analog domain on a physical level, resulting in the direct output of the relevant information $\alpha_i$ from the detector itself without any need for further external data processing.

## Photodetectors as spectral dot-product devices

At the heart of any optical spectroscopy application are photodetectors, devices that convert light into electrical signals. The most important parameter that governs the behavior of a photodetector is the photoresponsivity $R = I_{ph}/P$, the ratio between the generated photocurrent $I_{ph}$ (or photovoltage $V_{ph}$) and the incident light power $P$, expressed in A/W (or V/W). On closer look, however, $R$ is not just a constant, but a complicated function $R(\nu, \mathbf{J}, \mathbf{x}, \ldots)$ that may depend on a number of variables, such as the frequency of the incident light $\nu$, its polarization $\mathbf{J}$, the spatial position on the device $\mathbf{x}$, and several other parameters, including the power $P$ itself. While the usual engineering practice is to make $R$ as independent of these variables as possible (within a relevant parameter range), here we take the exact opposite approach. The spectral responsivity $R(\nu)$ of our photodetector is tailored to serve a specific purpose, and can be freely configured – or programmed – to perform any linear regression or classification task.

In the linear response regime, a photodetector with frequency-dependent photoresponsivity $R(\nu)$ illuminated with light of the spectrum $P(\nu)$ generates a photocurrent

$$I_{ph} = \int_\nu R(\nu)P(\nu)\,\mathrm{d}\nu = \langle R(\nu), P(\nu)\rangle = \mathbf{r}^\mathsf{T}\mathbf{p}. \tag{1}$$

If the spectrum and spectral responsivity are represented in a finite-dimensional vector space $\mathbb{R}^n$ by two column vectors[2] $\mathbf{p} = \left(P(\nu_1), P(\nu_2), \ldots, P(\nu_n)\right)^\mathsf{T}$ and $\mathbf{r} = \left(R(\nu_1), R(\nu_2), \ldots, R(\nu_n)\right)^\mathsf{T}$, respectively, the photoresponse $I_{ph}$ can be conveniently written as an inner product, as depicted in Eq. (1). By performing the above operation $m$

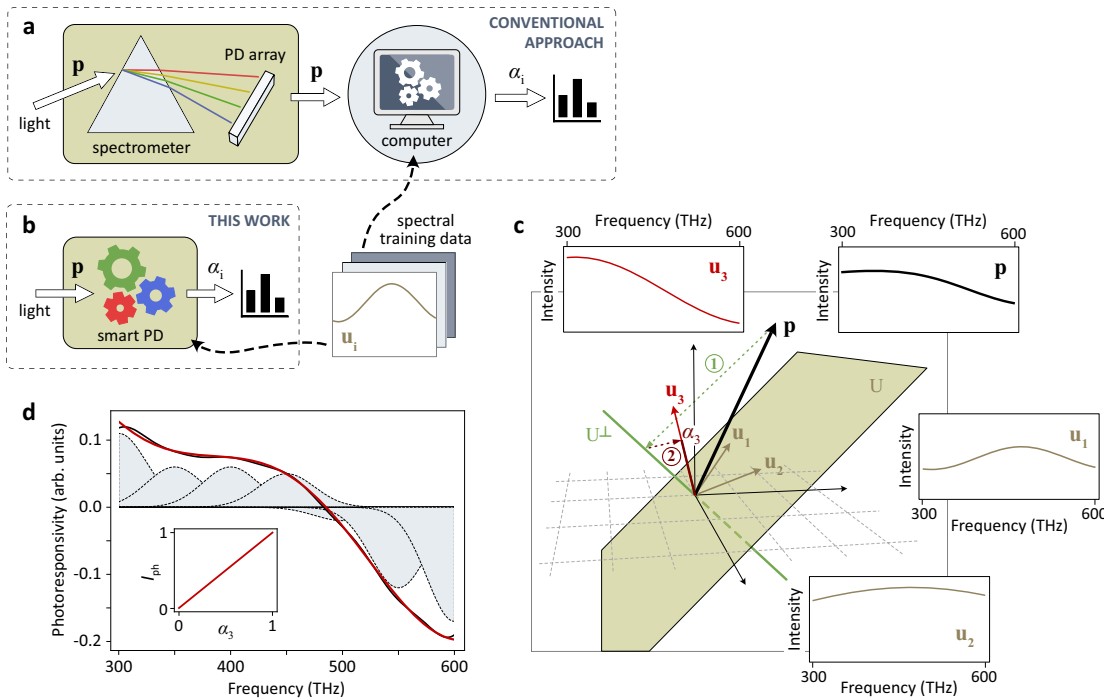

**Fig. 1 | Operational scheme.** Comparison between **a**, a conventional spectral sensing scheme and **b**, the smart photodetection approach presented in this article. While conventionally a spectrum $\mathbf{p}$ is processed externally to extract the desired coefficients $\alpha_i$, in our in-sensor computing scheme the same information is provided by the detector itself. $\mathbf{u}_i$ is a set of spectra used for training. PD, photodetector. **c**, Geometrical interpretation of the implemented algorithm. ① $\mathbf{p}$ is first projected onto the subspace $U^\perp$, and ② the outcome is then projected onto the spectral signature of interest $\mathbf{u}_m$. Note that while in this particular example, $U^\perp$ is a one-dimensional subspace (represented by the green line), it is generally high-dimensional. U (represented by the shaded plane in this example) is the subspace spanned by the unwanted spectral signatures. The inset in the upper right corner shows an optical spectrum that is a linear combination of three spectra (shown in the other insets), $\mathbf{p} = \alpha_1\mathbf{u}_1 + \alpha_2\mathbf{u}_2 + \alpha_3\mathbf{u}_3$, with coefficients $\alpha_1 = \alpha_2 = 0.25$ and $\alpha_3 = 0.5$. **d**, Calculated spectral responsivity vector $\mathbf{r}$ for a photodetector that is sensitive to $\mathbf{u}_3$ only (red line), and its approximation (black line) by a sum of $N = 7$ Gaussians (dashed lines). Inset: Simulated photodetector output (normalized) when $\alpha_3$ is varied between 0 and 1, while $\alpha_1$ and $\alpha_2$ are held constant. $I_{ph}$, photocurrent.

times involving different spectral responsivity vectors, a matrix-vector multiplication $\mathbf{Rp}$ with responsivity matrix $\mathbf{R} = (\mathbf{r}_1^T, \mathbf{r}_2^T, \ldots, \mathbf{r}_m^T)^T$ can also be carried out. Tailoring the spectral response of a photodetector thus allows for the implementation of any linear model on the inner product space $\mathbb{R}^n$. As with computational spectrometers, such a photodetector measures a weighted superposition of spectral intensities at different wavelengths, but with the goal of directly extracting a sought-after quantity rather than reconstructing the underlying spectrum. While a frequency-dependent response can also be achieved by using a customized optical filter in front of a conventional photodetector[21], such a simple approach lacks reconfigurability and the ability to achieve negative photoresponses (i.e., a reversed current flow direction) necessary for the implementation of most machine learning algorithms.

## Linear model implementation

Linear regression models have proven valuable in various fields, including remote sensing and optical spectroscopy. In remote sensing, linear models can be used to determine land cover, vegetation density, water content, and other environmental parameters. In optical spectroscopy they can be employed to provide quantification of analytes in chemical mixtures. The relationship between the spectral response and the characteristic of interest is modeled using a linear function, where the coefficients of the function represent the weights assigned to each variable. The coefficients are estimated from training data using statistical methods, such as least squares regression. The resulting model can then be used to predict the property of interest for a new sample based on a spectral measurement. The general equation for a linear regression model is

$$y = c + \beta_1 p_1 + \beta_2 p_2 + \ldots + \beta_n p_n + \varepsilon = c + \boldsymbol{\beta}^T \mathbf{p} + \varepsilon \tag{2}$$

where $y$ is the response variable (such as the concentration of an analyte or the physical property of a material), the vector $\mathbf{p} = (p_1, p_2, \ldots, p_n)^T$ contains the predictor variables (which in the case of optical spectroscopy and remote sensing are the spectral intensities at different wavelengths), the vector $\boldsymbol{\beta} = (\beta_1, \beta_2, \ldots, \beta_n)^T$ contains the regression coefficients (i.e. the weights assigned to each predictor variable), $c$ is an offset, and $\varepsilon$ is a random error term (the difference between the predicted and the true response). Based on the formal similarity between Eqs. (1) and (2), it can be concluded that linear models can be implemented in a photodetector's linear response by equating its spectral responsivity $\mathbf{r}$ with the regression vector $\boldsymbol{\beta}$. $c$ can be incorporated in the detector output by adding a constant offset current (in many cases it may also be set to zero).

## Analysis of spectral mixtures

As an illustrative example, we present the implementation of spectral mixture analysis[22–24], a powerful technique widely used to extract information from complex spectral data. In fluorescence microscopy, for example, this technique can be employed to separate the signals of different fluorophores with overlapping spectral profiles. It thus allows to obtain quantitative information about the distribution and abundance of multiple fluorophores used to label different cellular structures or molecules in biological imaging. Similarly, it can be employed in Raman spectroscopy to identify and quantify the different chemical components of a sample. The method works by decomposing an optical spectrum $\mathbf{p}$ into a linear combination of the spectra $\mathbf{u}_i$ of $m$ spectrally distinct components, $\mathbf{p} = \sum_{i=1}^m \alpha_i \mathbf{u}_i$, where the coefficients $\alpha_i$ reflect the proportions of the individual components in the mixture. Let $\mathbf{u}_m$ be the spectral signature of interest, and the remaining m-1 spectra are undesired signatures that we assemble into a matrix $\mathbf{U} = (\mathbf{u}_1 \mathbf{u}_2 \cdots \mathbf{u}_{m-1})$. As described in the Methods section, a regression operator for $\mathbf{u}_m$ can be derived by proceeding in two steps. First, one eliminates the effects of the undesired spectral contributions by

projecting $\mathbf{p}$ onto the orthogonal complement of the column space of $\mathbf{U}$. The result is then projected onto $\mathbf{u}_m$ to obtain an output that is directly proportional to $\alpha_m$. Figure 1c provides a geometrical interpretation of the algorithm. By comparison with Eq. (1) we can deduce the frequency-dependent photoresponsivity

$$\mathbf{r}^T = \mathbf{u}_m^T \left( \mathbf{I} - \mathbf{U} \left( \mathbf{U}^T \mathbf{U} \right)^{-1} \mathbf{U}^T \right). \tag{3}$$

A photodetector that possesses a spectral response according to Eq. (3) projects a measured optical spectrum $\mathbf{p}$ onto the spectral signature of interest $\mathbf{u}_m$, while simultaneously cancelling the contributions of the other classes. The larger the proportion of $\mathbf{u}_m$ in $\mathbf{p}$, the larger the output current. Equation (3) can be further extended to include constraints (such as e.g. the full additivity condition $\sum_{i=1}^m \alpha_i = 1$) that can likewise be implemented in our photodetector, provided they are linear.

To further illustrate the underlying concept, we show in the upper right inset in Fig. 1c an example of an optical spectrum $\mathbf{p}$ that is a linear superposition of three spectra $\mathbf{u}_1$, $\mathbf{u}_2$ and $\mathbf{u}_3$ (shown in the other insets). The shapes of the spectra are known, but not how much each of them contributes to the total. We are interested in determining the contribution of $\mathbf{u}_3$. To realize a photodetector that is sensitive to $\mathbf{u}_3$ only, we determine the required photoresponsivity vector $\mathbf{r}$ from Eq. (3) and plot the result as red line in Fig. 1d (the Python script is provided in Supplementary Note 1). The outcome is intuitive: $\mathbf{u}_3$ mainly contributes at lower frequencies, while $\mathbf{u}_1$ and $\mathbf{u}_2$ are spectrally broader and also have components at higher frequencies. Due to the progression of $\mathbf{r}$ from positive to negative values, the integrated contributions of $\mathbf{u}_1$ and $\mathbf{u}_2$ cancel out, while only $\mathbf{u}_3$ contributes to the net photocurrent $I_{ph}$. As shown in Fig. 1d (inset), $I_{ph}$ is indeed directly proportional to $\alpha_3$.

Other commonly used regression models in optical spectroscopy include partial least squares regression (PLS), multiple linear regression (MLR), and principal component regression (PCR). These methods differ from spectral mixture analysis only in the way the regression/responsivity vector is calculated, but can be implemented physically in the very same way. The implementation of linear classifiers is also feasible. Consider the general equation of a binary classifier $y = \sigma_c(\mathbf{w}^T \mathbf{p})$, where $\mathbf{w}$ is the weight vector, $\sigma_c$ represents a threshold function, and $\mathbf{p}$ is the input. By equating the spectral responsivity $\mathbf{r}$ with the weight vector $\mathbf{w}$, any binary linear classification algorithm can be realized. The specific shape of $\mathbf{r}$ depends on the chosen algorithm (perceptron, naïve Bayes, linear discriminant analysis (LDA), support vector machine (SVM), etc.). The threshold function can be implemented in hardware using external electronics connected to the output of the detector. The specific choice of the regression or classification method depends on the characteristics of the data and the problem at hand.

Sensors equipped with signal processing capabilities that allow them to perform advanced and sophisticated functions beyond the basic measurement and detection capabilities of traditional devices are referred to as[25] intelligent or smart. This is also the case here, as our photodetector is able to detect desired spectral features and disregard others. The calculation according to Eq. (2) is performed on the lowest possible hierarchy level – the physical level of photon detection – and does not require a spectrometer or any further signal processing to analyze the data. The benefits of such an approach are obvious. Reading out a single electrical signal requires less time, resources, and energy than recording and analyzing the entire spectrum. In fact, the energy consumption of the device is merely due to small leakage currents. The power consumption of the device presented below is estimated to be on the order of 10 nW, which represents a significant improvement over conventional solutions that typically consume at least tens of mW (e.g. Hamamatsu mini-spectrometer C12666A,

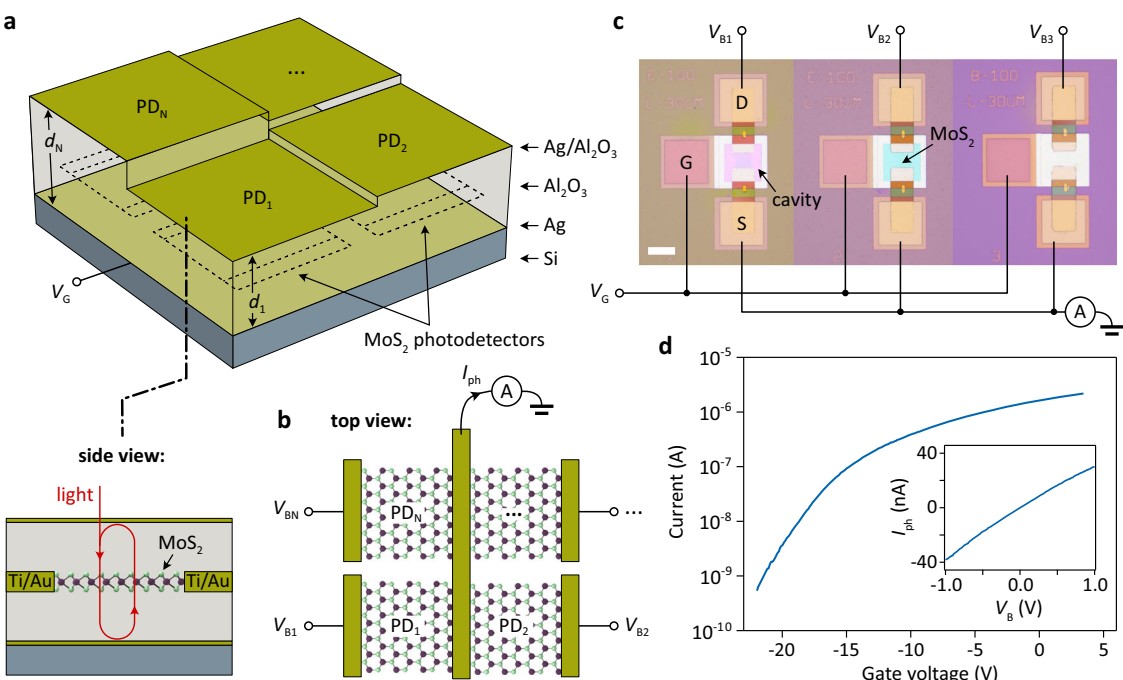

**Fig. 2 | Physical implementation using a 2D semiconductor. a** Schematic of the device layout. $MoS_2$ photodetectors $PD_i$, $i = 1...N$, are embedded in Fabry-Pérot cavities with different thicknesses $d_i$, $i = 1...N$, and resonance frequencies. The dashed lines schematically indicate the positions of the $MoS_2$ metal-semiconductor-metal detectors within the cavities. $V_G$, gate voltage. **b** The source electrodes are interconnected to sum the photocurrents generated by each segment. Each of the drain electrodes is supplied by a separate voltage $V_{Bi}$, $i = 1...N$, to set the responsivity of the respective segment. **c** Microscopic image of the fabricated device, consisting of $N = 3$ resonators, and interconnection of the individual segments. Scale bar, 50 μm. **d** Device current versus gate voltage characteristic for one of the segments; the other two segments show similar behavior and their characteristics are presented in Supplementary Fig. 1. Inset: The photocurrent/responsivity can be controlled by the applied bias voltage.

30 mW). The data analysis takes place in real-time and is constrained solely by the physics of the process of photocurrent generation. Furthermore, a large number of such detectors can potentially be arranged in a two-dimensional (2D) array to realize a smart imaging system that responds only to objects exhibiting a certain spectral signature.

## Prototype implementation using a 2D semiconductor

Our physical implementation of a photodetector with programmable spectral response is based on the use of $N$ optical microcavities with equally spaced center frequencies $\nu_i$, $i = 1...N$. Each cavity contains a metal-semiconductor-metal (MSM) photodetector whose response is strongly enhanced at the resonance frequency. The overall responsivity of such a device can be modelled by $R(\nu) \approx \sum_{i=1}^{N} a_i g(\nu - \nu_i)$, where $g(\nu)$ is a (Gaussian) line shape function originating from the resonator. The parameters $a_i$ correspond to the peak responsivities of the respective resonator segments and can be controlled by voltages applied to the MSM elements, allowing an approximation to a desired $R(\nu)$ as described in the Methods section. The result is shown in Fig. 1d as black line, where an excellent approximation of the desired responsivity is achieved.

Figure 2a depicts the actual device structure based on a chemical vapor deposition (CVD) grown 2D semiconductor film. For details regarding the fabrication process, refer to the Methods section. The $N$ optical resonators are formed by planar Fabry-Pérot microcavities consisting of a highly reflective metallic bottom mirror, a semi-transparent metallic top mirror, and $Al_2O_3$ buffer layers with different thicknesses $d_i$, $i = 1...N$. The resonance frequencies of the individual segments are given by $\nu_i = c_0/(2n_{Al2O3}d_i)$, where $c_0$ is the velocity of light in vacuum and $n_{Al2O3}$ is the refractive index of $Al_2O_3$. Chemical vapor deposition (CVD)-grown $MoS_2$ MSM photodetectors are embedded between the mirrors. A 2D semiconductor was chosen as the active material because of its excellent optoelectronic

properties[26,27], ease of integration into an optical cavity[28–30], and ease of electrical control of its properties using both gate and bias voltages. The source electrodes are connected together to sum the photocurrents generated by each segment, as shown in Fig. 2b. Each of the drain electrodes is supplied by a separate voltage $V_i$, $i = 1...N$, to set the photoresponsivity of the respective segment. This allows the programming of an arbitrary photoresponsivity vector $\mathbf{r}$. The bottom mirror also serves as a gate electrode to control the carrier concentrations in the $MoS_2$ layer (Fig. 2d). A negative gate bias ($V_G = -35\,V$) was applied in all measurements to fully deplete the $MoS_2$ channel from carriers and suppress dark current (see Supplementary Fig. 2a). As previously reported[31–33], the photoresponse of $MoS_2$ photodetectors originates from photogating due to a charge transfer into trap states in the $MoS_2$ layer itself and/or its surrounding and shows an approximately linear behavior with bias voltage $V_B$ as depicted in Fig. 2d, inset. The peak responsivity at $V_G = -35\,V$ and $V_B = 1\,V$ amounts to -0.5 A/W. Within the investigated intensity range, the photocurrent demonstrates a linear behavior, with no trap state filling observed (see Supplementary Fig. 2b). This observation validates the applicability of the theory presented above, which relies on the assumption of a linear response.

## Programmable spectral response and spectroscopy demonstration

In Fig. 2c we present a microscope image of our prototype device consisting of $N = 3$ segments with nominal cavity lengths of $d_1 = 0.12$ μm, $d_2 = 0.14$ μm and $d_3 = 0.16$ μm. The corresponding reflectance spectra, depicted in Fig. 3a, confirm the resonant absorption at the design wavelengths. Figure 3b shows the photoresponsivities of each of the three segments. Compared to the reflectance spectra, the photocurrent resonances are broadened and slightly blue-shifted. The broadening is due to the cavity's inhomogeneity caused by the metallic

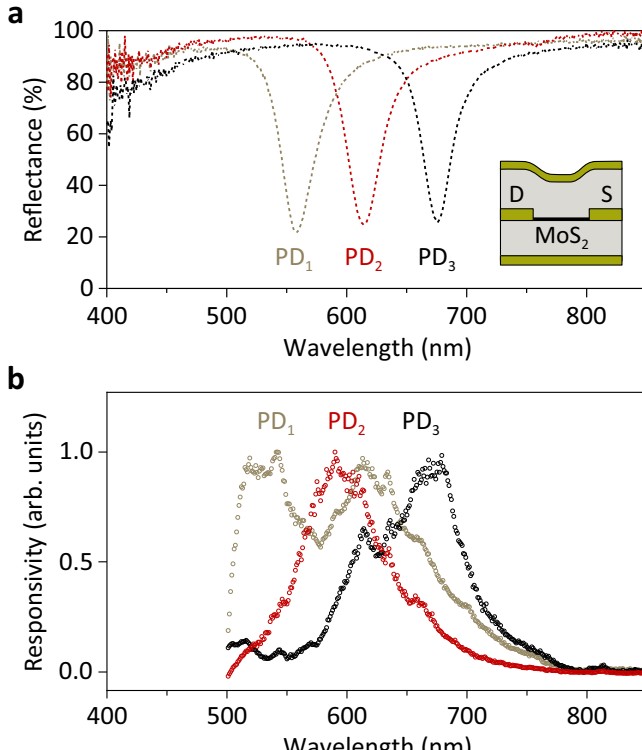

**Fig. 3 | Resonant cavity-enhanced MoS₂ photodetectors. a** Reflectance spectra of the three MoS₂ photodetector segments. Inset: Schematic of the device cross-section illustrating the inhomogeneity of the resonator near the metal contacts. D, drain electrode; S, source electrode. **b** Normalized spectral responsivities for each of the three segments.

contacts with finite height (while the reflectance is obtained from a μm-sized spot in the center of the device, the photocurrent is generated over the entire channel, particularly in the vicinity of the contacts where inhomogeneity is strongest), while the blue shift is attributed to the increase in MoS₂ absorption at smaller wavelengths[34,35]. One detector segment exhibits an additional spurious resonance at around 620 nm, likely due to device imperfections and absorption enhancement from the B-exciton resonance. In practice, however, the additional resonance does not significantly affect the functionality of the sensor as it can be compensated by the other two detector segments.

Despite consisting of only three wavelength selective elements, our prototype MoS₂ photodetector exhibits a certain degree of versatility and can be effectively used in diverse spectroscopy applications within the visible regime. For instance, as shown in the first three graphs of Fig. 4a, the device can potentially approximate the emission spectra of widely-used fluorescent dyes that function as tracers in biology and medicine. The shaded areas in the graphs represent the targeted spectra, while the lines depict the spectral responsivities that can be set through the specific combination and tuning of the three elements. Furthermore, the remaining graphs in Fig. 4a demonstrate the device's versatility by showcasing the approximation of several generic spectra, some of which exhibit negative response.

Our current device realization is limited by the large bandgap of MoS₂, which restricts its response to wavelengths below ~700 nm. Using graphene in place of MoS₂ could expand the device's response to longer wavelengths, particularly into the spectroscopically important infrared region[36,37]. The number of resonators required for optimal performance depends on the field of application. For fluorescence spectroscopy of biological samples, for example, we estimate that a device with about ten wavelength-selective elements (e.g., resonators)

would be versatile enough to fully cover the visible spectral range (380–700 nm), considering that the dyes used in this type of application have spectral linewidths that are typically greater than 30 nm (see also Fig. 4a).

To experimentally verify the smart sensing capabilities of our photodetector, we used a white light source along with a set of optical transmission filters to generate spectra **u₁**, **u₂** and **p**, as shown in Fig. 4b. The intensities of **u₁** and **u₂** were independently adjusted as described in the Methods section. The spectral response **r** required to detect the contribution of **u₂** and simultaneously cancel **u₁** was determined from Eq. (2) and is plotted as black line in Fig. 4c. By scaling the segment responsivities with the factors $a_1 = 0.88$, $a_2 = 0.0$ and $a_3 = -1.0$, we can approximately set the desired spectral response in our device (red line in Fig. 4c). Experimentally, this is achieved by applying bias voltages of $V_{B1} = 0.7$ V, $V_{B2} = 0.0$ V and $V_{B3} = -1.0$ V. If we vary now the intensity of **u₂** while keeping **u₁** constant, the photocurrent $I_{ph}$ increases linearly with optical power as expected (Fig. 4d). On the other hand, if we vary **u₁** while keeping **u₂** constant, no change in output is observed (Fig. 4e). The photocurrent stems solely from the spectral contribution **u₂**.

## Discussion

In summary, we presented a spectroscopy scheme in which the spectral response of a photodetector is programmed such that the device is able to perform a specific regression or classification task. We used a prototype consisting of three resonant cavity-enhanced MoS₂ photodetectors with individually adjustable peak responsivities to determine the fraction of a component in a spectral mixture. The underlying concept can be applied to any spectroscopic algorithm that can be represented by a linear operator. The current limitations of the device, such as the limited spectral range and deviations from Gaussian spectral line shapes due to the large bandgap of MoS₂ and its strongly wavelength-dependent absorption spectrum, respectively, can be overcome, for example, by using graphene as the active photodetection material.

In addition to cavity-enhanced devices, there are several other physical mechanisms that enable a wavelength-dependent photo-response. One such mechanism involves utilizing plasmonic resonances. By engineering the geometry and composition of nanostructures, it is possible to achieve strong interactions with specific wavelengths of light, thereby enhancing the photoresponse at those wavelengths[38]. Another avenue for achieving a wavelength-dependent photoresponse is through the use of materials with inherent wavelength-dependent properties. For example, organic semiconductors and perovskite materials can be tailored to have specific energy band gaps, allowing them to selectively absorb and generate charge carriers in response to different wavelengths of light[39,40]. By harnessing these physical mechanisms, it should be possible to realize photodetectors with a tailored wavelength-dependent photoresponse.

Finally, our concept is not limited to spectroscopy alone, but can in the future also be applied to smart imaging systems. Current hyperspectral imagers[41] consist of an array of photosensors and a spectrometer. Both the spectral and spatial domains are combined to form a multi-dimensional data set that, after data processing, is often reduced to a single number representation per pixel, such as the concentration of a particular chemical compound in a 2D map. The devices are bulky, their price is currently prohibitive for cost-effective routine applications, and their size prevents their use in modern portable/wearable devices. To overcome these drawbacks, we propose to process multispectral information in a 2D array of smart detectors, presented here. Such an image sensor in combination with a tailored algorithm can provide spatially resolved information for a specific measure with high speed, low energy consumption and high sensitivity. With its ability to provide real-time analysis and interpretation of spectroscopic signals, it has the potential to be applied in a wide range

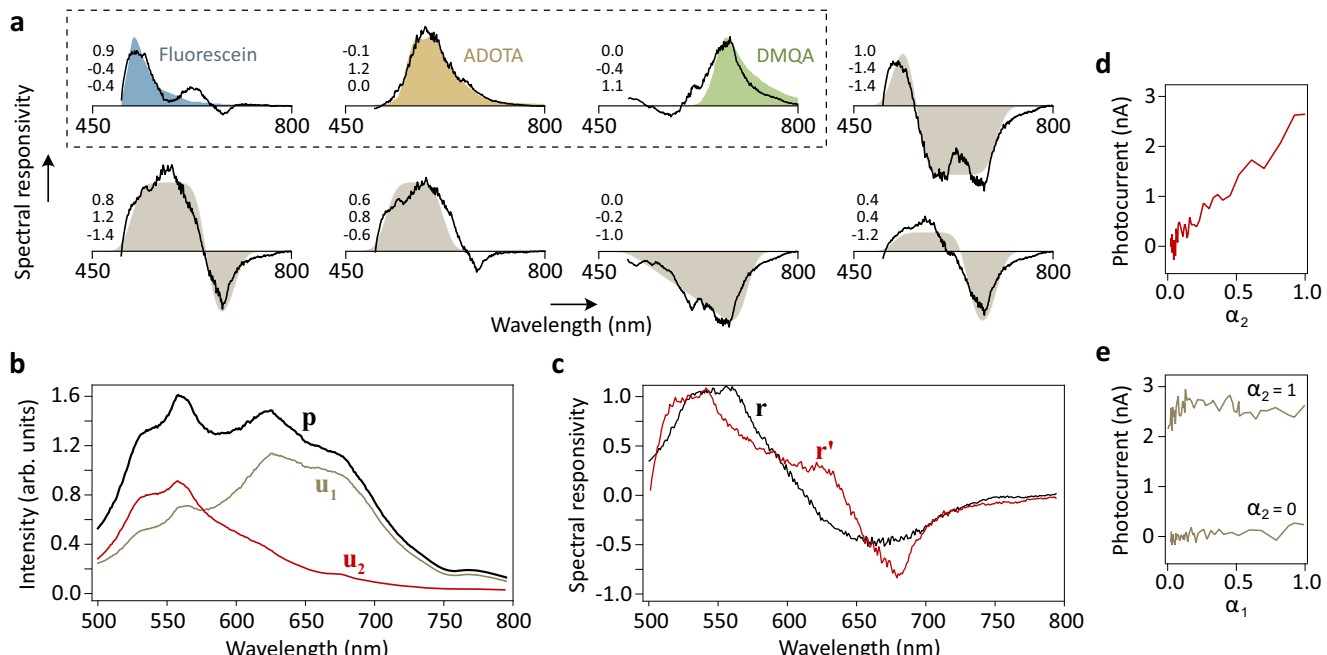

**Fig. 4 | Programmable spectral response and spectroscopy demonstration. a** Exemplary spectral responsivities. The shaded areas represent the desired responses, while the corresponding lines depict the best achievable least-squares approximations obtained through linear combinations of the spectral responsivities in Fig. 3b. The numbers indicate the contributions $a_i$ of the individual detector segments. The first three spectra in the figures correspond to widely-used fluorescent dyes (fluorescein, azadioxatriangulenium (ADOTA), dimethylquinacridone (DMQA)), while the remaining spectra illustrate some generic examples. **b** Optical spectra $\mathbf{p} = \alpha_1\mathbf{u}_1 + \alpha_2\mathbf{u}_2$ used in experiments, shown here for $\alpha_1 = \alpha_2 = 1$. **c** Optimal spectral responsivity $\mathbf{r}$ (black line) and its approximation $\mathbf{r}'$ in our photodetector (red line). **d** $I_{ph}$ when $\alpha_2$ is varied between 0 and 1, and $\alpha_1 = 1$ is kept constant. **e,** $I_{ph}$ when $\alpha_1$ is varied between 0 and 1, and $\alpha_2 = 0$ or 1.

of industries and scientific fields, from industrial process control to medical diagnostics.

# Methods

## Spectral mixture analysis

Consider a mixture of $m$ spectrally distinct components, represented by their respective spectra $\mathbf{u}_i \in \mathbb{R}^n$, and let $\mathbf{p} \in \mathbb{R}^n$ be a linear combination of these spectra with coefficients $\alpha_i \in \mathbb{R}$,

$$\mathbf{p} = \sum_{i=1}^{m} \alpha_i \mathbf{u}_i. \qquad (4)$$

To obtain a linear operator for a specific component of interest, say $\mathbf{u}_m$, it is necessary to eliminate the effects of interfering signatures represented by the columns of matrix $\mathbf{U}$, given by $\mathbf{U} = (\mathbf{u}_1 \mathbf{u}_2 \cdots \mathbf{u}_{m-1})$. This can be done by projecting $\mathbf{p}$ onto a subspace that is orthogonal to the columns of $\mathbf{U}$, resulting in a vector that contains energy only associated with $\mathbf{u}_m$ and Gaussian noise $\mathcal{N}(0, \sigma_n^2)$. Therefore, we first apply the orthogonal subspace projection (OSP) operator[22]

$$\mathbf{P} = (\mathbf{I} - \mathbf{U}\mathbf{U}^+), \qquad (5)$$

where $\mathbf{U}^+$ is the pseudo-inverse of $\mathbf{U}$, given by $\mathbf{U}^+ = (\mathbf{U}^T\mathbf{U})^{-1}\mathbf{U}^T$, and $\mathbf{I}$ is the identity matrix. We then apply the operator $\boldsymbol{\xi}^T$ that maximizes the signal-to-noise-ratio

$$\text{SNR} = \frac{\alpha_m^2}{\sigma_n^2} \frac{\boldsymbol{\xi}^T \mathbf{P}_\perp \mathbf{u}_m \mathbf{u}_m^T \mathbf{P}^T \boldsymbol{\xi}}{\boldsymbol{\xi}^T \mathbf{P}_\perp \mathbf{P}_\perp^T \boldsymbol{\xi}}. \qquad (6)$$

As shown in ref. 22, the operator that maximizes this quotient is $\boldsymbol{\xi}^T = \kappa \mathbf{u}_m^T$, with $\kappa$ being an arbitrary proportionality factor. Altogether, this leads to the regression vector $\boldsymbol{\xi}^T \mathbf{P}$, and after substitution for $\boldsymbol{\xi}^T$ and $\mathbf{P}$, we ultimately arrive at Eq. (3).

## Approximation to R(ν)

The parameters $a_i$ required to approximate a desired $R(\nu)$ through a sum of shifted Gaussians $\sum_{i=1}^{N} a_i g(\nu - \nu_i)$ can be determined from the linear system $\mathbf{Ga} = \mathbf{r}$, where $\mathbf{r} = (R(\nu_1), \ldots, R(\nu_N))^T$ is a vector containing the desired responsivity values at the support points $\nu_i$ as calculated from Eq. (3), $\mathbf{G} = (g_{ki}) \in \mathbb{R}^{N \times N}$ is a matrix with elements $g_{ki} = (\sqrt{2\pi}\sigma)^{-1} \exp(-0.5(\nu_k - \nu_i)^2/\sigma^2)$ and $\mathbf{a} = (a_1, \ldots, a_N)^T$.

## Device fabrication

The cavity in the resonant photodetectors is formed by silver mirrors separated from the CVD-grown $MoS_2$ film by aluminium oxide ($Al_2O_3$) layers. The sample was fabricated by the following procedure. All lithography steps were performed by electron-beam lithography using a Raith eLINE system and Allresist PMMA 679.004. The metal deposition was carried out in a Plassys MEB550SL electron-beam evaporation system at a pressure of $3$–$5 \times 10^{-8}$ mbar. The fabrication starts with the patterning of the Ti/Au gate pads with the subsequent patterning of the bottom resonator mirror, made out of 100 nm thick silver. An 80-nm-thick $Al_2O_3$ gate oxide was then deposited using atomic layer deposition. A $MoS_2$ film was grown by CVD on sapphire[42] and then transferred onto the target wafer[43]. The grown film is continuous over an area of >50 mm² with monolayer thickness and small multi-layer $MoS_2$ islands. In the next step, rectangular $MoS_2$ channels were patterned and subsequently etched using Ar/SF6 plasma. The drain and source contacts were then formed by another EBL process followed by Ti/Au (3/30 nm) deposition. Additional patches were added on the edges of the bottom metal mirror and contact pads to assure the integrity of the electrical contact. The channels were then covered by 80-nm-thick $Al_2O_3$ gate oxide which was subsequently etched by wet chemical etching in potassium hydroxide at specific places to achieve the resonators with desired cavity lengths. In the next step, the semi-transparent upper resonator mirror was patterned and evaporated using nominally 30 nm of silver and 2 nm of aluminium which serves as

a passivation layer to protect silver from oxidation. Finally, the oxide on the contact pads was etched to open the contact pads for wire bonding.

## Experimental setup

The optical measurements were carried out with an optical setup, consisting of two beam paths with two different light sources and a lens to focus the light onto the device. Measurements were performed using either an NKT SuperK compact source with an acousto-optic tuneable filter for acquiring the wavelength-dependent photo-responsivities or a white-light source OSL1 from Thorlabs with a broad spectrum in the visible range. To generate light with optical spectrum **p**, we split the OSL1 output into two paths using a beam splitter, employed filters from Edmund Optics (46543) and Thorlabs (FL632.8-3) in both paths to create the two spectra $\mathbf{u}_1$ and $\mathbf{u}_2$, respectively, and recombined the outputs in a second beam splitter. Note that limiting the experiment to only two spectra does not result in any loss of generality. Continuously tuneable neutral density filters with transparency varying from 0.04 to 2.0 OD were used in both paths to individually adjust the $\alpha_1$ and $\alpha_2$ contributions. Keithley 2612B source measurement units were used to set the bias voltages and record the photocurrent. Prior to the optical measurements, the electrical performance of the device was verified in a Lakeshore TTPX sample station connected to an Agilent HP 4155 C semiconductor parameter analyzer.

## Data availability

The data that support the findings of this study are available from the corresponding author upon request.

## Code availability

The code and algorithm in this paper are available from the corresponding author upon request.

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

## Acknowledgements
We thank Juraj Darmo for technical assistance and acknowledge financial support by the European Union (grant agreement No. 785219 Graphene Flagship).

## Author contributions
T.M. conceived the device concept and supervised the project. D.K. and D.K.P. fabricated the device, built the experimental setup, carried out the measurements and analyzed the data. T.M. and D.K.P. prepared the manuscript. All authors discussed the results and commented on the manuscript.

## Competing interests
The authors declare no competing interests.
