## [Peer Review File · Nature Communications]

In-sensor computing using a MoS₂ photodetector with programmable spectral responsivityREVIEWER COMMENTS

Reviewer #1 (Remarks to the Author):

In this paper, the authors described a very interesting "smart" spectroscopic photodetector. This photodetector aims not to record or analyze the spectrum, but to selectively measure the intensity of the spectral signature of interest. The idea is novel and interesting. However, I think the authors should reorganize the manuscript to improve the readability.

First, the present manuscript organization is logic-oriented. This is fine to describe a simple story. However, I think the story in this manuscript is too complexed, covering broad range of knowledge. Considering the readership of the manuscript, I strongly recommend the authors to use mission-oriented organization. For example, they could raise remote sensing or other spectrum-related image treatment as their mission. An understandable mission is better for conserving the audience's patience than the lengthy algorithm. For example, although the definition of u_i is described, I still think it is still not easy for general readers because the article is not mission-oriented.

Next, the introduction of the algorithm itself can be simplified. I think a more understandable description is possible. Meanwhile I suggest to move all the calculations to derive Eq. 2 to the SI while keep Eq. 2 itself in the main text, so as the core story flow is highlighted.

Third, the description of G , v_i , the photogating effect and the blue shift seems too lengthy and not tightly related to the story. The relationship between the linear superposition of u_i and the device performance should be strengthened. In the present version, these two parts are too isolated. When this relationship is clarified and confirmed, the authors can solidly prove their device as "programmable".

Finally, additional data is necessary to demonstrate its usage in image processing. Preferably, the authors should quantitatively measure the reduction in calculation load by using their photodetector as imaging array pixel. As declared by the authors, their detector should be useful in a mission that extracts certain information about a specimen under examination from an optical image. Once this mission is demonstrated, it would be very helpful to improve the readability. If it is too difficult to realize a real array, a simulated demonstration would be helpful.

Minor suggestions are as follows.

1. The conditions to realize the spectral responsivities in Figure 4a should be provided.
2. In the abstract, "The device can be programmed to cover different types of applications" should be corrected to something like "...optical feature classification tasks"
3. I kind of disagree with the authors' definition that "Sensors equipped with signal processing capabilities to suppress the influence of unwanted variables at low levels" are "intelligent" or "smart" because a simple RC circuit is able to filter certain signals. Maybe programmable or reconfigurable is better to define "intelligent" or "smart".

Reviewer #2 (Remarks to the Author):

In this manuscript, the authors reported the design and realization of programmable spectral response photodetectors. Based on the use of several optical microcavities with equally spaced centre frequencies, the photodetector can be programmed to cover different types of applications. Taking the analysis of spectral mixtures as an example, they first reveal the mathematical principle of the analysis, and then the prototype implementation is produced using MoS₂ photodetectors with adjustable responsivities. The idea is new and could be of great interest to the community. On the downside, there are some weaknesses that need to be addressed before publication:

- 1) The four insets in Fig.1b are inconsistent, two of them are missing the vertical axis title.
- 2) In Fig.4c, where is the unit of spectral responsivity?
- 3) According to the authors, the scheme can be applied to any other algorithm that can be represented by a linear operator, it is better to present more examples to support the superiority of the scheme.
- 4) The authors can present more about the properties of a MoS₂ photodetector in the supplementary material.
- 5) It is better to compare the in-sensor computing method to the traditional method in the article.

- 6) The prototype device consists of only three segments, this is not enough to convince that the device can form any spectral responsivities. It is better to apply more segments into the device and show more random spectral responsivities they form. This is very important, please do provide the new results in the revised manuscript.
- 7) The twelfth reference paper is not about computational spectrometers, wrong citation.

Reviewer #3 (Remarks to the Author):

Dear Authors,

I think this paper is suggesting an in-sensor computation that can be expanded to any linear operations. I think the paper is well written and easy to follow and I have no problem following the work from the beginning to the end. And I have one question and one comment.

The authors mention about the limitation of current device you provided and that can be solved as changing PD materials. And I am wondering if this conceptual device can really solve any linear operations as long as PD is changed with Graphene. I mean I am wondering what the actual limitations are due to the number of PD's or Aluminum oxide thickness control, the control of the shape of responsivity and so on.

Also, I am wondering if the authors can provide a little complex example than two variable case. I understand that the system can be expanded into more complex systems. But actual realization is very important. And I think sharing the issues with more complex system will help the community much more than the current version.

Author's response to referees' comments and questions

Referee #1:

In this paper, the authors described a very interesting "smart" spectroscopic photodetector. This photodetector aims not to record or analyze the spectrum, but to selectively measure the intensity of the spectral signature of interest. The idea is novel and interesting. However, I think the authors should reorganize the manuscript to improve the readability.

We would like to extend our sincere appreciation to the referee for his/her valuable suggestions and feedback provided during the review process. We deeply appreciate the time and dedication invested in thoroughly evaluating the manuscript. Moreover, we are very grateful for the referee's recognition of the novelty of the work. In response to the referee's comments, we have implemented substantial revisions and enhancements to the manuscript, which are highlighted in red. Below, we provide a comprehensive response to the referee's comments and questions, preceded by the respective bolded quotations.

First, the present manuscript organization is logic-oriented. This is fine to describe a simple story. However, I think the story in this manuscript is too complexed, covering broad range of knowledge. Considering the readership of the manuscript, I strongly recommend the authors to use mission-oriented organization. For example, they could raise remote sensing or other spectrum-related image treatment as their mission. An understandable mission is better for conserving the audience's patience than the lengthy algorithm. For example, although the definition of ui is described, I still think it is still not easy for general readers because the article is not mission-oriented.

We thank the referee for this comment and in response have made very significant changes to improve the accessibility of the paper. We have relegated the hard-to-digest spectral unmixing algorithm to the background, and instead resort now to a more easily comprehensible general linear regression model. In particular, we show that any linear regression model can be implemented in photodetector hardware by equating the responsivity vector of the device to the regression vector of the model. We show the same for linear classifiers. Importantly, this change not only increases the comprehensibility of the manuscript, but also expands the scope and significance of the work. In response to the referee's suggestion, we have further included fluorescence spectroscopy as an illustrative application example that is a common thread throughout the manuscript and provides a practical context for the reader. We believe that by introducing an illustrative example, without diminishing the breadth of the work, the paper will appeal to a broader audience. In detail, we have implemented the following changes:

- The referee raises a valid criticism regarding the accessibility of the presented algorithm to a broader audience. To address this concern, we have moved the algorithm to the background (see also next comment) and expanded the scope of the work. Our approach

now begins with a general introduction of linear regression models, which we explain in a clear and easily understandable manner. We demonstrate that any linear model can be effectively implemented by equating the responsivity vector to the regression vector:

“**Linear model implementation**

Linear regression models have proven valuable in various fields including remote sensing and optical spectroscopy. In remote sensing, for example, linear models can be used to determine land cover, vegetation density, water content, and other environmental parameters. In optical spectroscopy they can be employed to provide quantification of analytes in chemical mixtures. The relationship between the spectral response and the characteristic of interest is modeled using a linear function, where the coefficients of the function represent the weights assigned to each variable. The coefficients are estimated from training data using statistical methods such as least squares regression. The resulting model can then be used to predict the property of interest for new samples based on their spectral properties. The general equation for a linear regression model is

$$y = c + \beta_1 p_1 + \beta_2 p_2 + \dots + \beta_n p_n + \varepsilon = c + \boldsymbol{\beta}^T \mathbf{p} + \varepsilon \quad (2)$$

where y is the response variable (such as the concentration of an analyte or the physical property of a material), the vector $\mathbf{p} = (p_1, p_2, \dots, p_n)^T$ contains the predictor variables (which in the case of optical spectroscopy and remote sensing are the spectral intensities at different wavelengths), the vector $\boldsymbol{\beta} = (\beta_1, \beta_2, \dots, \beta_n)^T$ contains the regression coefficients (i.e. the weights assigned to each predictor variable), c is an offset, and ε is a random error term (the difference between the predicted and the true response). Based on the formal similarity between Eqs. (1) and (2), it can be concluded that linear models can be implemented in a photodetector's linear response by equating its spectral responsivity \mathbf{r} with the regression vector $\boldsymbol{\beta}$. c can be incorporated in the detector output by adding a constant offset current; in many cases it may also be set to zero.”

- In the second paragraph we show that this is also true for any linear *classifier*, namely by choosing the responsivity vector according to the weight vector and applying a threshold at the detector output:

“Other commonly used regression models in optical spectroscopy include partial least squares regression (PLS), multiple linear regression (MLR), and principal component regression (PCR). These methods differ from spectral mixture analysis only in the way the regression/responsivity vector is calculated, but can be implemented physically in the very same way. The implementation of linear classifiers is also feasible. Consider the general equation of a binary classifier $y = \sigma_c(\mathbf{w}^T \mathbf{p})$, where \mathbf{w} is the weight vector, σ_c represents a threshold function, and \mathbf{p} is the input. By equating the spectral responsivity \mathbf{r} with the weight vector \mathbf{w} , any binary linear classification algorithm can be realized. The specific shape of \mathbf{r} depends on the chosen algorithm (perceptron, naïve Bayes, linear discriminant analysis (LDA), support vector machine (SVM), etc.). The threshold function can be implemented in hardware using external electronics connected to the output of the detector.

The specific choice of the regression or classification method depends on the characteristics of the data and the problem at hand.”

- Only then we do address the spectral unmixing algorithm as an example. Furthermore, in response to the referee's suggestion, we have incorporated fluorescence spectroscopy as an illustrative application example:

“Analysis of spectral mixtures

As an illustrative example, we present the implementation of spectral mixture analysis, a powerful technique widely used to extract information from complex spectral data. In fluorescence microscopy, for example, this technique can be employed to separate the signals of different fluorophores with overlapping spectral profiles. It thus allows to obtain quantitative information about the distribution and abundance of multiple fluorophores used to label different cellular structures or molecules in biological imaging. Similarly, it can be employed in Raman spectroscopy to identify and quantify the different chemical components of a sample.”

- The example of fluorescence spectroscopy serves as a consistent thread that runs throughout the manuscript. By integrating this example as a recurring motif, we establish a coherent and cohesive narrative that enhances understanding. It resurfaces in Figure 4a:

“For instance, as shown in the first three graphs of Figure 4a, the device can potentially approximate the emission spectra of widely-used fluorescent dyes that function as tracers in biology and medicine. The shaded areas in the graphs represent the targeted spectra, while the lines depict the spectral responsivities that can be set through the specific combination and tuning of the three elements.”

The inclusion of the spectroscopy example provides practical context and real-world relevance, capturing the interest of readers from diverse backgrounds. Simultaneously, the detailed explanations of linear models and their relation to our device implementation ensure a thorough understanding of the underlying principles and methodologies. We believe that by striking a balance between practical application and theoretical foundations, our work fosters a deeper appreciation of our contributions and resonates with a wider range of readers.

Next, the introduction of the algorithm itself can be simplified. I think a more understandable description is possible. Meanwhile I suggest to move all the calculations to derive Eq. 2 to the SI while keep Eq. 2 itself in the main text, so as the core story flow is highlighted

We agree with the referee's assessment. In fact, especially with the newly added introduction of linear models (see above), it is not necessarily important for the reader to understand the specific details of the implemented algorithm. We hence...

- ...shortened the explanation of the algorithm in the main text to a few lines:
“The approach works by decomposing an optical spectrum \mathbf{p} into a linear combination of the spectra \mathbf{u}_i of m spectrally distinct components, $\mathbf{p} = \sum_{i=1}^m \alpha_i \mathbf{u}_i$, where the coefficients α_i

reflect the proportions of the individual components in the mixture. Let \mathbf{u}_m be the spectral signature of interest, and the remaining $m - 1$ spectra are undesired signatures that we assemble into a matrix $\mathbf{U} = (\mathbf{u}_1 \ \mathbf{u}_2 \ \dots \ \mathbf{u}_{m-1})$. As described in the Methods section, a regression operator for \mathbf{u}_m can be derived by proceeding in two steps. First, one eliminates the effects of the undesired spectral contributions by projecting \mathbf{p} onto the orthogonal complement of the column space of \mathbf{U} . The result is then projected onto \mathbf{u}_m to obtain an output that is directly proportional to α_m ."

- ...and moved the derivation of the algorithm to the Methods section, which also allowed us to expand its derivation:

"METHODS

Spectral mixture analysis. Consider a mixture of m spectrally distinct components, represented by their respective spectra $\mathbf{u}_i \in \mathbb{R}^n$, and let $\mathbf{p} \in \mathbb{R}^n$ be a linear combination of these spectra with coefficients $\alpha_i \in \mathbb{R}$,

$$\mathbf{p} = \sum_{i=1}^m \alpha_i \mathbf{u}_i. \quad (4)$$

To obtain a linear operator for a specific component of interest, say \mathbf{u}_m , it is necessary to eliminate the effects of interfering signatures represented by the columns of matrix \mathbf{U} , given by $\mathbf{U} = (\mathbf{u}_1 \ \mathbf{u}_2 \ \dots \ \mathbf{u}_{m-1})$. This can be done by projecting \mathbf{p} onto a subspace that is orthogonal to the columns of \mathbf{U} , resulting in a vector that contains energy only associated with \mathbf{u}_m and Gaussian noise $\mathcal{N}(0, \sigma_n^2)$. Therefore, we first apply the orthogonal subspace projection (OSP) operator^{Fehler! Textmarke nicht definiert.}

$$\mathbf{P}_\perp = (\mathbf{I} - \mathbf{U}\mathbf{U}^+), \quad (5)$$

where \mathbf{U}^+ is the pseudo-inverse of \mathbf{U} , given by $\mathbf{U}^+ = (\mathbf{U}^T\mathbf{U})^{-1}\mathbf{U}^T$, and \mathbf{I} is the identity matrix. We then apply the operator $\boldsymbol{\xi}^T$ that maximizes the signal-to-noise-ratio

$$\text{SNR} = \frac{\alpha_m^2 \boldsymbol{\xi}^T \mathbf{P}_\perp \mathbf{u}_m \mathbf{u}_m^T \mathbf{P}_\perp \boldsymbol{\xi}}{\sigma_n^2 \boldsymbol{\xi}^T \mathbf{P}_\perp \mathbf{P}_\perp^T \boldsymbol{\xi}}. \quad (6)$$

As shown in Ref. Fehler! Textmarke nicht definiert., the operator that maximizes this quotient is $\boldsymbol{\xi}^T = \kappa \mathbf{u}_m^T$, with κ being an arbitrary proportionality factor. Altogether, this leads to the regression vector $\boldsymbol{\xi}^T \mathbf{P}_\perp$, and after substitution for $\boldsymbol{\xi}^T$ and \mathbf{P}_\perp , we ultimately arrive at Eq. (3)."

Third, the description of G, vi, the photogating effect and the blue shift seems too lengthy and not tightly related to the story. The relationship between the linear superposition of ui and the device performance should be strengthened. In the present version, these two parts are to isolated. When this relationship is clarified and confirmed, the authors can solidly prove their device as "programmable".

Also here we followed the suggestions of the referee:

- We moved the derivation and description of $G_{a=r}$ to the Methods section:
“Approximation to $R(v)$. The parameters a_i required to approximate a desired $R(v)$ through a sum of shifted Gaussians $\sum_{i=1}^N a_i g(v - v_i)$ can be determined from the linear system $\mathbf{G}\mathbf{a} = \mathbf{r}$, where $\mathbf{r} = (R(v_1), \dots, R(v_N))^T$ is a vector containing the desired responsivity values at the support points v_i as calculated from Eq. (3), $\mathbf{G} = (g_{ki}) \in \mathbb{R}^{N \times N}$ is a matrix with elements $g_{ki} = (\sqrt{2\pi}\sigma)^{-1} \exp(-0.5 (v_k - v_i)^2 / \sigma^2)$ and $\mathbf{a} = (a_1, \dots, a_N)^T$.”
- The description of $G_{a=r}$ in the main text has been shortened accordingly:
 “...allowing an approximation to a desired $R(v)$ as described in the Methods section.”
- We have condensed the description of the broadening and blue-shift effects. By including a few concise lines, we aim to preemptively address potential questions that may arise among the readers:
“Compared to the reflectance spectra, the photocurrent resonances are broadened and slightly blue-shifted. The broadening is due to the cavity's inhomogeneity caused by the metallic contacts with finite height (while the reflectance is obtained from a μm -sized spot in the center of the device, the photocurrent is generated over the entire channel, particularly in the vicinity of the contacts where inhomogeneity is strongest), while the blue shift is attributed to the increase in MoS₂ absorption at smaller wavelengths. One detector segment exhibits an additional spurious resonance at around 620 nm, likely due to device imperfections and absorption enhancement from the B-exciton resonance. In practice, however, the additional resonance does not significantly affect the functionality of the sensor and can be compensated by the other two detector segments.”

Regarding the strengthening of the *“relationship between the linear superposition of u_i and the device performance”*, we would like to refer to our previous comments where we discuss the connection between linear models and the obtained regression/weight vector, which can be equated with the responsivity vector. By establishing this relationship, we have ensured coherency throughout the manuscript, effectively bridging the gap between the device and the underlying basic regression theory. This connection not only enhances the overall coherence of our work but also strengthens the understanding of how the device operates in accordance with the theoretical framework.

On the experimental side, we have now added a new Supplementary Figure 2b. This addition is crucial as it demonstrates a linear relationship between the output current of the device and the optical input power, which is essential for supporting the existing linear theory. In the main text we have added a corresponding reference to the Figure:

“Within the investigated intensity range, the photocurrent demonstrates a linear behavior, with no trap state filling observed (see Supplementary Figure S2b). This observation validates the applicability of the theory presented above, which relies on the assumption of a linear response.”

Finally, additional data is necessary to demonstrate its usage in image processing. Preferably, the authors should quantitatively measure the reduction in calculation load by using their

photodetector as imaging array pixel. As declared by the authors, their detector should be useful in a mission that extracts certain information about a specimen under examination from an optical image. Once this mission is demonstrated, it would be very helpful to improve the readability. If it is too difficult to realize a real array, a simulated demonstration would be helpful.

We do suggest in the manuscript a potential application of our smart detectors in imaging applications. But please note that this is an outlook for future investigations, while the focus of the current manuscript is on spectroscopy. As correctly mentioned by the referee, building a whole sensor array is difficult within the short period of a paper review process. (Let us note that we have submitted a collaborative 4-years project proposal to the European Commission for the realization of just such a system.) In the revised version of the manuscript, however, we made a quantitative comparison of our device with a conventional spectroscopy solution. In particular, we measured the energy consumption of our device and compare it to a commercial device:

“The power consumption of the device presented below is estimated to be on the order of 10 nW, which represents a significant improvement over conventional solutions that typically consume at least tens of mW (e.g. Hamamatsu mini-spectrometer C12666A, 30 mW). The data analysis takes place in real-time and is constrained solely by the physics of the process of photocurrent generation. Furthermore, a large number of such detectors can potentially be arranged in a two-dimensional (2D) array to realize a smart imaging system that responds only to objects exhibiting a certain spectral signature.”

Note that the 6 orders-of-magnitude improvement does not even take into account the power consumption of the microcontroller that reads the data from the spectrometer and runs the regression algorithm.

Minor suggestions are as follows.

1. The conditions to realize the spectral responsivities in Figure 4a should be provided.

We thank the referee for pointing out the absence of this information. We have added it to the figure, as well as a description in the figure caption.

2. In the abstract, “The device can be programmed to cover different types of applications” should be corrected to something like “...optical feature classification tasks”

We changed the text to: “The device can be programmed to cover different types of **spectral regression or classification tasks.**”

3. I kind of disagree with the authors' definition that "Sensors equipped with signal processing capabilities to suppress the influence of unwanted variables at low levels" are "intelligent" or "smart" because a simple RC circuit is able to filter certain signals. Maybe programmable or reconfigurable is better to define "intelligent" or "smart".

Good point. We changed the text to: "Sensors equipped with signal processing capabilities that allow them to perform advanced and sophisticated functions beyond the basic measurement and detection capabilities of traditional devices are referred to as "intelligent" or "smart"."

Referee #2:

In this manuscript, the authors reported the design and realization of programmable spectral response photodetectors. Based on the use of several optical microcavities with equally spaced centre frequencies, the photodetector can be programmed to cover different types of applications. Taking the analysis of spectral mixtures as an example, they first reveal the mathematical principle of the analysis, and then the prototype implementation is produced using MoS₂ photodetectors with adjustable responsivities. The idea is new and could be of great interest to the community. On the downside, there are some weaknesses that need to be addressed before publication:

We wish to express our gratitude to the referee for his/her invaluable suggestions and feedback during the review process. The time and dedication devoted to thoroughly evaluating the manuscript are deeply appreciated. Additionally, we are thankful for the referee's acknowledgment of the innovative nature of our work. In light of the referee's comments, we have diligently incorporated significant revisions and improvements into the manuscript, marked in red. Subsequently, we present a comprehensive response to the referee's comments and questions, preceded by the corresponding bolded quotations.

1) The four insets in Fig.1b are inconsistent, two of them are missing the vertical axis title.

We thank the referee for pointing out the inconsistency. We have made the necessary revisions.

2) In Fig.4c, where is the unit of spectral responsivity?

We added the information regarding the peak responsivity to the text:

"The peak responsivity at $V_G = -35$ V and $V_B = 1$ V amounts to ~ 0.5 A/W."

3) According to the authors, the scheme can be applied to any other algorithm that can be represented by a linear operator, it is better to present more examples to support the superiority of the scheme.

We thank the referee for this comment. In order to show that indeed *any linear (regression or classification) model* can be implemented using the concept presented in this work we have substantially revised the manuscript. In particular, we show now that any linear regression model can be implemented by equating the responsivity vector of the device to the regression vector of the model. We show the same for linear classifiers. Importantly, this change not only increases the comprehensibility of the manuscript, but also expands the scope and significance of the work:

“Linear model implementation

Linear regression models have proven valuable in various fields including remote sensing and optical spectroscopy. In remote sensing, for example, linear models can be used to determine land cover, vegetation density, water content, and other environmental parameters. In optical spectroscopy they can be employed to provide quantification of analytes in chemical mixtures. The relationship between the spectral response and the characteristic of interest is modeled using a linear function, where the coefficients of the function represent the weights assigned to each variable. The coefficients are estimated from training data using statistical methods such as least squares regression. The resulting model can then be used to predict the property of interest for new samples based on their spectral properties. The general equation for a linear regression model is

$$y = c + \beta_1 p_1 + \beta_2 p_2 + \dots + \beta_n p_n + \varepsilon = c + \boldsymbol{\beta}^T \mathbf{p} + \varepsilon \quad (2)$$

where y is the response variable (such as the concentration of an analyte or the physical property of a material), the vector $\mathbf{p} = (p_1, p_2, \dots, p_n)^T$ contains the predictor variables (which in the case of optical spectroscopy and remote sensing are the spectral intensities at different wavelengths), the vector $\boldsymbol{\beta} = (\beta_1, \beta_2, \dots, \beta_n)^T$ contains the regression coefficients (i.e. the weights assigned to each predictor variable), c is an offset, and ε is a random error term (the difference between the predicted and the true response). Based on the formal similarity between Eqs. (1) and (2), it can be concluded that linear models can be implemented in a photodetector's linear response by equating its spectral responsivity \mathbf{r} with the regression vector $\boldsymbol{\beta}$. c can be incorporated in the detector output by adding a constant offset current; in many cases it may also be set to zero.”

Besides the spectral unmixing algorithm described in detail in the manuscript, we also mention now a few other examples:

„Other commonly used regression models in optical spectroscopy include partial least squares regression (PLS), multiple linear regression (MLR), and principal component regression (PCR). These methods differ from spectral mixture analysis only in the way the regression/responsivity vector is calculated, but can be implemented physically in the very same way. “

We also discuss how a linear *classifier* can be implemented, namely by choosing the responsivity vector according to the weight vector and applying a threshold at the detector output:

„The implementation of linear classifiers is also feasible. Consider the general equation of a binary classifier $y = \sigma_c(\mathbf{w}^T \mathbf{p})$, where \mathbf{w} is the weight vector, σ_c represents a threshold function, and \mathbf{p} is the input. By equating the spectral responsivity \mathbf{r} with the weight vector \mathbf{w} , any binary linear classification algorithm can be realized. The specific shape of \mathbf{r} depends on the chosen algorithm (perceptron, naïve Bayes, linear discriminant analysis (LDA), support vector machine (SVM), etc.). The threshold function can be implemented in hardware using external electronics connected to the output of the detector. The specific choice of the regression or classification method depends on the characteristics of the data and the problem at hand.“

4) The authors can present more about the properties of a MoS2 photodetector in the supplementary material.

In response, we have made an addition to the manuscript by including the power-dependence of the optical response in Supplementary Figure S2b. This new figure, along with Supplementary Figure S1 (gate-voltage dependence) and Figure 2d (bias-voltage dependence), Figure 3b (spectral response), and Supplementary Figure S2a (responsivity versus gate-voltage and dark current), collectively provide all relevant aspects of the photodetector performance within the scope of this work.

5) It is better to compare the in-sensor computing method to the traditional method in the article.

In response to the referee’s comment, we have implemented two improvements in the manuscript. Firstly, we recognized that the original illustration in Figure 1a lacked clarity. Therefore, we have revised the illustration to ensure it effectively communicates our findings. Additionally, we have expanded the accompanying text to provide more comprehensive explanations and enhance the reader's understanding of the figure:

“In Figures 1, we compare the detection scheme presented in this article to a conventional **optical** sensing approach. Conventionally (**Figure 1a**), incident light is dispersed to produce a spectrum and its corresponding vector representation, \mathbf{p} , is recorded. **The digitized data are then processed using an external processing unit/computer, which utilizes a statistical or machine-learning algorithm to analyze \mathbf{p} and extract the desired information, α_i , from the measurement.** In contrast, in our **in-sensor computing** scheme (**Figure 1b**), the **same** algorithm is implemented in the analog domain on a physical level, resulting in the direct output of the relevant information α_i from the detector **itself without any need for further external data processing.**”

Furthermore, we have conducted a comparison of the energy consumption between a conventional spectroscopy system and our proposed approach. This comparison allows us to highlight the advantages and potential benefits of our method over traditional systems:

“The power consumption of the device presented below is estimated to be on the order of 10 nW, which represents a significant improvement over conventional solutions that typically consume at least tens of mW (e.g. Hamamatsu mini-spectrometer C12666A, 30 mW). The data analysis takes place in real-time and is constrained solely by the physics of the process of photocurrent generation.”

6) The prototype device consists of only three segments, this is not enough to convince that the device can form any spectral responsivities. It is better to apply more segments into the device and show more random spectral responsivities they form. This is very important, please do provide the new results in the revised manuscript.

We acknowledge and understand the concern regarding the limitations of our experimental demonstration, but please note that in practice there will always be limitations in terms of wavelength range, spectral resolution, etc. and any device can only be designed for a certain range of applications. Optical transitions in biological samples, for example, are spectrally rather broad (typically tens of nm), but their optical fingerprints extend over a wide spectral range. In Raman spectroscopy, on the other hand, transitions are spectrally narrow but occur only over a narrow range. In our particular case, the relatively large bandgap of MoS₂ (1.85 eV, or 670 nm) and the limited tuning range of our light-source (>500 nm) limit our experiments to only a part of the visible spectrum. Nevertheless, we demonstrate in a revised Figure 4a the ability of our photodetector to approximate the emission spectra of commonly used fluorescence markers in biology. A potential application there lies in mitigating the unwanted fluorescence emissions originating from the “wrong” markers. We acknowledge that employing graphene as a replacement for MoS₂ would allow to extend the spectral range (but also requires a more complex device concept to avoid dark currents). Despite the restricted spectral coverage, we strongly believe that the importance of this work remains undiminished and our prototype device demonstrates a sufficient level of versatility for a proof-of-concept. We believe that the work showcases the potential of our novel spectroscopy approach and provides a foundation for future developments and optimizations to achieve broader coverage in different spectroscopy applications.

To address these issues and provide an outlook we added (besides the revised Figure 4a) the following two paragraphs to the manuscript:

“Despite consisting of only three wavelength selective elements, our prototype MoS₂ photodetector exhibits a certain degree of versatility and can be effectively used in diverse spectroscopy applications within the visible regime. For instance, as shown in the first three graphs of Figure 4a, the device can potentially approximate the emission spectra of widely-used fluorescent dyes that function as tracers in biology and medicine. The shaded areas in

the graphs represent the targeted spectra, while the lines depict the spectral responsivities that can be set through the specific combination and tuning of the three elements. Furthermore, the remaining graphs in Figure 4a demonstrate the device's versatility by showcasing the approximation of several generic spectra, some of which exhibit negative response."

"Our current device realization is limited by the large bandgap of MoS₂, which restricts its response to wavelengths below ~700 nm. Using graphene in place of MoS₂ could expand the device's response to longer wavelengths, particularly into the spectroscopically important infrared region. The number of resonators required for optimal performance depends on the field of application. For fluorescence spectroscopy of biological samples, for example, we estimate that a device with about ten wavelength-selective elements (e.g., resonators) would be versatile enough to fully cover the visible spectral range (380-700 nm), considering that the dyes used in this type of application have spectral linewidths that are typically greater than 30 nm (see also Figure 4a)."

7) The twelfth reference paper is not about computational spectrometers, wrong citation.

We thank the referee for pointing out this error. We have removed the citation.

Referee #3:

Dear Authors,

I think this paper is suggesting an in-sensor computation that can be expanded to any linear operations. I think the paper is well written and easy to follow and I have no problem following the work from the beginning to the end. And I have one question and one comment.

We would like to thank the referee for his/her invaluable suggestions and feedback. We deeply appreciate the time and effort dedicated to thoroughly evaluate the manuscript. We are pleased to hear that the referee found the paper well written and easy to follow. Taking into account the referee's comments, we have diligently implemented substantial revisions and improvements in the manuscript, which are highlighted in red. Below, we provide a comprehensive response to the referee's comments and questions, preceded by the respective bolded quotations.

The authors mention about the limitation of current device you provided and that can be solved as changing PD materials. And I am wondering if this conceptual device can really solve any linear operations as long as PD is changed with Graphene. I mean I am wondering what the actual limitations are due to the number of PD's or Aluminum oxide thickness control, the control of the shape of responsivity and so on.

The answer to this question has two parts. The first part concerns the question whether, in principle, *any linear (regression or classification) operator/model* can be implemented using the concept of a photodetector with a wavelength-dependent responsivity. To answer this question, we have added the following two paragraphs to the manuscript. In the first paragraph we clearly show that any linear *regression* model can be implemented by equating the spectral responsivity vector of a photodetector with the regression vector of the linear model:

“Linear model implementation

Linear regression models have proven valuable in various fields including remote sensing and optical spectroscopy. In remote sensing, for example, linear models can be used to determine land cover, vegetation density, water content, and other environmental parameters. In optical spectroscopy they can be employed to provide quantification of analytes in chemical mixtures. The relationship between the spectral response and the characteristic of interest is modeled using a linear function, where the coefficients of the function represent the weights assigned to each variable. The coefficients are estimated from training data using statistical methods such as least squares regression. The resulting model can then be used to predict the property of interest for new samples based on their spectral properties. The general equation for a linear regression model is

$$y = c + \beta_1 p_1 + \beta_2 p_2 + \dots + \beta_n p_n + \varepsilon = c + \boldsymbol{\beta}^T \mathbf{p} + \varepsilon \quad (2)$$

where y is the response variable (such as the concentration of an analyte or the physical property of a material), the vector $\mathbf{p} = (p_1, p_2, \dots, p_n)^T$ contains the predictor variables (which in the case of optical spectroscopy and remote sensing are the spectral intensities at different wavelengths), the vector $\boldsymbol{\beta} = (\beta_1, \beta_2, \dots, \beta_n)^T$ contains the regression coefficients (i.e. the weights assigned to each predictor variable), c is an offset, and ε is a random error term (the difference between the predicted and the true response). Based on the formal similarity between Eqs. (1) and (2), it can be concluded that linear models can be implemented in a photodetector's linear response by equating its spectral responsivity \mathbf{r} with the regression vector $\boldsymbol{\beta}$. c can be incorporated in the detector output by adding a constant offset current; in many cases it may also be set to zero.”

In the second paragraph we show that this is also true for any linear *classifier*, namely by choosing the responsivity vector according to the weight vector and applying a threshold at the detector output:

“Other commonly used regression models in optical spectroscopy include partial least squares regression (PLS), multiple linear regression (MLR), and principal component regression (PCR). These methods differ from spectral mixture analysis only in the way the regression/responsivity vector is calculated, but can be implemented physically in the very same way. The implementation of linear classifiers is also feasible. Consider the general equation of a binary classifier $y = \sigma_c(\mathbf{w}^T \mathbf{p})$, where \mathbf{w} is the weight vector, σ_c represents a threshold function, and \mathbf{p} is the input. By equating the spectral responsivity \mathbf{r} with the weight

vector \mathbf{w} , any binary linear classification algorithm can be realized. The specific shape of \mathbf{r} depends on the chosen algorithm (perceptron, naïve Bayes, linear discriminant analysis (LDA), support vector machine (SVM), etc.). The threshold function can be implemented in hardware using external electronics connected to the output of the detector. The specific choice of the regression or classification method depends on the characteristics of the data and the problem at hand.”

The second part of our answer to the referee’s question concerns the concrete physical realization in a device. As mentioned by the referee, graphene indeed seems to be a good material for this type of application due to its wavelength-independent absorption over a wide spectral range. Graphene also has a relatively low optical absorption of 2% which makes it easier to achieve narrower linewidths. The reason why we chose MoS₂ in this work is that there the responsivity can be more easily adjusted via a bias voltage. If a graphene detector were used, it would have to be operated in PTE mode (to avoid dark current), which complicates the device geometry but nevertheless is feasible.

The specifics of the device depends on the type of application. In the manuscript we now discuss fluorescence spectroscopy in biology as an application example. The fluorophores used in this field typically emit in the range 400-700 nm, which can be covered with the materials used in our device (including the chosen dielectric). Fluorescence linewidths are typically broader than 30 nm due to thermal and other broadening mechanisms. Approximately 10 resonator elements would hence be sufficient to fully cover the whole visible spectral range. In a previous publication (*Nano Lett.* 2012, 12, 2773), we demonstrated microcavity-integrated graphene photodetectors with a linewidth down to 9 nm; the 30 nm are hence easily achievable. We have added this information to the manuscript:

“Our current device realization is limited by the large bandgap of MoS₂, which restricts its response to wavelengths below ~700 nm. Using graphene in place of MoS₂ could expand the device's response to longer wavelengths, particularly into the spectroscopically important infrared region. The number of resonators required for optimal performance depends on the field of application. For fluorescence spectroscopy of biological samples, for example, we estimate that a device with about ten wavelength-selective elements (e.g., resonators) would be versatile enough to fully cover the visible spectral range (380-700 nm), considering that the dyes used in this type of application have spectral linewidths that are typically greater than 30 nm (see also Figure 4a).”

However, many fluorophores have a linewidth that is spectrally broader, rather around 60-70 nm. We can therefore approximate their spectral shapes reasonably well even with our device, consisting of only three segments. We present the results in the revised Figure 4a. Furthermore, the use of microcavity-enhanced devices is only one way to achieve a wavelength-dependent responsivity. In the manuscript, we therefore now also provide an outlook on other possible realizations:

“In addition to cavity-enhanced devices, there are several other physical mechanisms that enable a wavelength-dependent photoresponse. One such mechanism involves utilizing

plasmonic resonances. By engineering the geometry and composition of nanostructures, it is possible to achieve strong interactions with specific wavelengths of light, thereby enhancing the photoresponse at those wavelengths. Another avenue for achieving a wavelength-dependent photoresponse is through the use of materials with inherent wavelength-dependent properties. For example, organic semiconductors and perovskite materials can be tailored to have specific energy band gaps, allowing them to selectively absorb and generate charge carriers in response to different wavelengths of light. By harnessing these physical mechanisms, it should be possible to realize photodetectors with a tailored wavelength-dependent photoresponse.”

Also, I am wondering if the authors can provide a little complex example than two variable case. I understand that the system can be expanded into more complex systems. But actual realization is very important. And I think sharing the issues with more complex system will help the community much more than the current version.

We thank the referee for this comment. Through our experimental demonstrations, we have showcased the device's ability to reject an unwanted spectrum \mathbf{u}_1 by configuring its spectral responsivity \mathbf{r} orthogonally to \mathbf{u}_1 , i.e. $\mathbf{P}_\perp \mathbf{u}_1 = 0$ (lower line in Figure 4e). This rejection capability remains intact even in the presence of other light \mathbf{u}_2 , as demonstrated by the upper line in Figure 4e. Therefore, our experimental results validate the expected device behavior.

The inclusion of additional spectra, let's say a third spectrum \mathbf{u}_0 , in the experiment does not yield any additional insights. In such a scenario, the algorithm would give an \mathbf{r} orthogonal to both \mathbf{u}_0 and \mathbf{u}_1 , i.e. $\mathbf{P}_\perp \mathbf{u}_0 = 0$ and $\mathbf{P}_\perp \mathbf{u}_1 = 0$. Consequently, it would also be orthogonal to any linear combination of \mathbf{u}_0 and \mathbf{u}_1 , which can be interpreted as a new spectrum $\mathbf{u}'_1 = \alpha_0 \mathbf{u}_0 + \alpha_1 \mathbf{u}_1$ where $\mathbf{P}_\perp \mathbf{u}'_1 = 0$. But this is exactly the scenario that we just discussed above and that we have validated experimentally. Limiting ourselves to only two spectra does hence not result in any loss of generality. We briefly mention this now in the manuscript:

“Note that limiting the experiment to only two spectra does not result in any loss of generality.”

There is also a simple practical reason: with each additional beam path (spectrum) we add to the setup, we lose an order of magnitude in light intensity, resulting in a worse signal-to-noise ratio.

REVIEWERS' COMMENTS

Reviewer #1 (Remarks to the Author):

In the revised manuscript, the authors addressed my concerns satisfactorily. I therefore suggest accepting this manuscript. However, most of the fonts in the figures are too small while the labels in Figure 4a are in very light colors. Please improve this in the next version.

Reviewer #2 (Remarks to the Author):

The authors have answered all the questions I had, it is a nice piece of work combining the programmable spectral response with In-sensor computing based on 2D semiconductors, and is therefore recommended for publishing in Nature Communications.

Reviewer #3 (Remarks to the Author):

Dear Authors,

I appreciate your efforts on adding examples and details. I have no further questions or requests. Thanks.

Author's response to referees' comments and questions

Referee #1:

In the revised manuscript, the authors addressed my concerns satisfactorily. I therefore suggest accepting this manuscript. However, most of the fonts in the figures are too small while the labels in Figure 4a are in very light colors. Please improve this in the next version.

We would like to extend our sincere appreciation to the referee for his/her valuable suggestions and feedback provided during the review process. We deeply appreciate the time and dedication invested in thoroughly evaluating the manuscript.

In response to the referee's suggestions regarding the figures, we implemented the following revisions: (i) We increased the font size in Figures 1b-d, 2a-d, 3a-b, and 4a by ~10%. (ii) We made the text colors in Figure 4a darker to improve readability.

Referee #2:

The authors have answered all the questions I had, it is a nice piece of work combining the programmable spectral response with In-sensor computing based on 2D semiconductors, and is therefore recommended for publishing in Nature Communications.

We wish to express our gratitude to the referee for his/her invaluable suggestions and feedback during the review process. The time and dedication devoted to thoroughly evaluating the manuscript are deeply appreciated.

Referee #3:

Dear Authors,

I appreciate your efforts on adding examples and details. I have no further questions or requests.

Thanks.

We would like to thank the referee for his/her invaluable suggestions and feedback. We deeply appreciate the time and effort dedicated to thoroughly evaluate the manuscript.